# VIPER: VIBRANT PERIOD REPRESENTATION FOR ROBUST AND EFFICIENT TIME SERIES FORECASTING

## ABSTRACT

In a data-driven world teeming with vast volumes of time series data, forecasting models play a pivotal role. The real-world time series data often exhibits intricate periodic patterns and trends, posing challenges for accurate modeling. Existing methods, reliant on fixed parameters and sampling techniques, may struggle to capture these complexities effectively. This paper designs a Vibrant Period Representation Enrichment (VIPER) framework, which effectively and dynamically harnesses the inherent multi-periodic nature of time series data. The VIPER framework adeptly separates the input sequence into trend and seasonal components. A Temporal Aggregation Block is specifically deployed for processing the seasonal component, applying innovative multi-period transformations compounded with a global self-attention mechanism. This configuration enables a comprehensive capture of both short-term and long-term period information, culminating in a vibrant period representation true to the essence of the temporal dynamics. Remarkably, experimental results from eight different time series forecasting datasets substantiate the superior performance, simplicity, and computational efficiency of VIPER compared with the state-of-the-art. Furthermore, VIPER has highlighted the advantages of employing longer input sequences, addressing the well-known Input Length Bottleneck Problem.

## 1  INTRODUCTION

Time series data pervades our data-driven world, with applications ranging from traffic flow estimation (Lv et al., 2014), energy management (Zhou et al., 2022a), disease control (Li et al., 2023), to financial investment (Lai et al., 2018). Time series forecasting (TSF), which relies on historical data, is a long-established task that has witnessed a substantial evolution in solution methodologies over the past few decades (Torres et al., 2021; Lim & Zohren, 2021). Real-world time series data often exhibits periodic patterns, such as supermarkets' weekly, monthly, and yearly sales fluctuations. These overlapping and interacting periods introduce complexity into modelling variations. Each time point is not only influenced by its immediate temporal pattern but also strongly connected to variations in adjacent periods, categorized as intraperiod-variation (short-term patterns within a period) and interperiod-variation (long-term trends across different periods). The evolution to deal with the TSF task has seen the transition from traditional statistical techniques (Geurts et al., 1977) to machine learning methods (Hochreiter & Schmidhuber, 1997; Cho et al., 2014) and deep learning-based approaches (Zhou et al., 2022a; Li et al., 2019; Zhang & Yan, 2023).

Recent research endeavours have focused on enhancing time series forecasting by obtaining resilient representations capable of effectively capturing essential patterns such as trends and seasonality while remaining robust to noise. These efforts employing fixed parameters have included techniques such as using a moving average kernel to decompose the sequence into its seasonal and trend components (Wu et al., 2021; Zhou et al., 2022b; Zeng et al., 2023), or segmenting the sequence into a series of overlapping or non-overlapping patches based on patch length (Nie et al., 2023). Alternatively, some methods exploit the inherent characteristic that the original sequence retains much of its vital information after sampling (LIU et al., 2022). For example, LightTS (Zhang et al., 2022) utilizes interval and continuous sampling techniques to divide the input sequence into more semantically informative tokens, while SCINet (LIU et al., 2022) adopts a "downsample-convolve-interact" architecture, sampling the input sequence following a binary tree structure to generate odd and even sub-sequences. Nevertheless, the utilization of fixed-parameter methods as described above

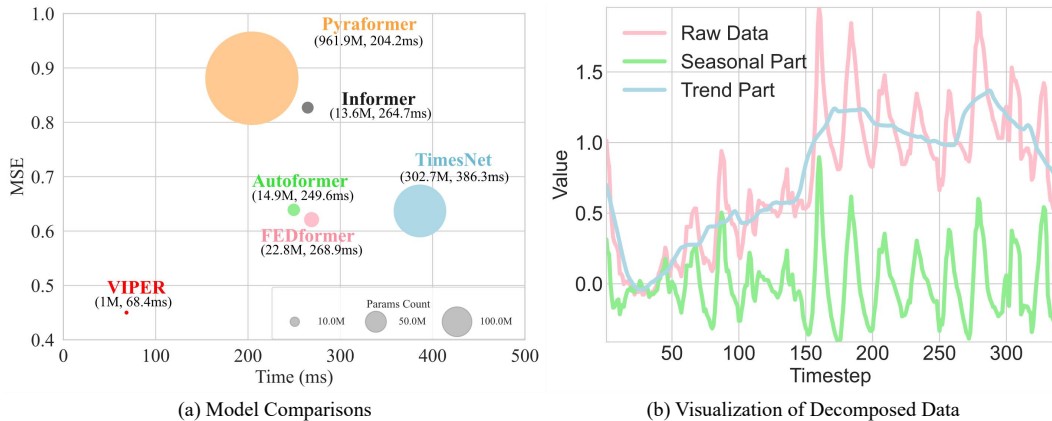

(a) Model Comparisons

(b) Visualization of Decomposed Data

Figure 1: (a) Comparison among different methods with respect to performance, parameters, and efficiency on Traffic dataset (Wu et al., 2021). VIPER showcases top-tier performance, minimal parameters, and the swiftest processing speed. (b) Data visualization example before and after data decomposition. After decomposition, the interference from the trend component is eliminated, and the seasonal part exhibits a more pronounced periodicity compared to the original input sequence.

brings complexities to simultaneously representing these distinct variations. Such fixed parameters or sampling strategies are often ill-suited for the varying and overlapping periodic patterns commonly encountered in real-world time series data. Consequently, they may not effectively capture the complexities of intraperiod-variation and interperiod-variation, as they impose rigid structures on the data that may not align with the dynamic nature of time series. Furthermore, most mainstream transformer-based models often suffer from a redundancy of parameters, leading to expensive training costs and long inference time, making them challenging to align with real-world applications, as illustrated in Figure 1 (a).

The above discrepancy motivated us to explore a novel approach, leveraging the multi-periodic characteristics inherent in real-world time series data to improve the feature representation. This concept was initially introduced by TimesNet (Wu et al., 2023), which transforms 1D time series into 2D representations using multiple period information and applies 2D convolutional kernels to handle these representations. However, directly applying such processing to the raw input sequence may fail to reveal the nuances of intraperiod-variation and interperiod-variation. A time series consists of both seasonal and trend components, with the former exhibiting repetitive patterns or periodicity and the latter capturing short and long-term variations, which is demonstrated in Figure 1 (b). The trend component blurs the periodic changes in the short term by reducing fluctuations within a period, making intraperiod-variation challenging to identify. Simultaneously, it also increases the consistency in fluctuations between different periods, making interperiod-variation hard to detect, as it typically obscures the differences between periods. Consequently, the trend component helps to complicate the disentanglement of intraperiod-variation and interperiod-variation from the raw input sequence. Moreover, due to the local self-attention mechanism of convolutional kernels, long-term period information is simply overlooked (Wu et al., 2023). Moreover, TimesNet fails to address the issue of parameter redundancy, the blue circle as shown in Figure 1 (a).

To more effectively disentangle and harness intraperiod-variation and interperiod-variation within time series data, we introduce the Vibrant Period Representation Enrichment (VIPER) framework. Our framework initiates by eliminating the influence of the trend component through decomposition and subsequently employs multi-period transformations on the seasonal component. Furthermore, it employs a global self-attention mechanism, in contrast to 2D convolutional kernels, to model the resulting 2D tensor. This strategy enables us to effectively capture both short-term and long-term period information concurrently.

We conducted experiments on eight widely used real-world datasets, and the experimental results demonstrate that our method consistently achieves state-of-the-art results across almost all of these datasets. Moreover, as indicated in Figure 1 (a), our method exhibits exceptional performance in

terms of model parameter count, inference time, and prediction accuracy, with a linear layer as the backbone. Our contributions are summarized as follows.

- We introduce VIPER, a versatile time series forecasting framework designed to harness both inter-period and intra-period relationships within input sequences.

- Through experiments conducted on eight public datasets, we illustrate that VIPER achieves superior performance compared to current state-of-the-art models. Remarkably, this is accomplished using only two basic linear layers, emphasizing simplicity and efficiency.

- As a versatile framework, our VIPER can seamlessly integrate with all major Time Series Forecasting (TSF) models and significantly enhance their predictive performance. Moreover, VIPER highlights the advantages of employing longer input sequences, addressing the Input Length Bottleneck Problem.

## 2 RELATED WORK

Owing to the real-world demand for time series prediction, several endeavours have been undertaken in this area. The first to tackle this challenge were statistical models, such as ARIMA (Geurts et al., 1977). However, as a recursive model, the predictive performance of ARIMA may deteriorate with an increase in the forecasting horizon, known as the "error accumulation effect" making long-term forecasts unstable. Additionally, ARIMA models assume that time series data is linear, implying that they struggle to effectively capture complex nonlinear relationships and features.

Under the machine learning and deep learning scenarios, there have been many remarkable achievements, primarily falling into the following four domains. In the domain of Recurrent Neural Networks (RNNs), classic designs (Hochreiter & Schmidhuber, 1997; Cho et al., 2014) remain widely employed and continue to hold relevance today. However, RNNs tend to process information in the same way across all time scales, which means they may perform poorly when dealing with patterns at different time scales. Within Convolutional Neural Networks (CNNs), introducing a unique "downsample-convolve-interact" architecture also achieves promising results (LIU et al., 2022). But CNNs are primarily designed to capture local patterns, which means they are better suited for addressing issues related to certain local features within time series data. For problems involving long-range dependencies, CNNs may be less effective.

Transformers (Vaswani et al., 2017) renowned for their proficiency in capturing semantic relationships within extensive sequences have catalyzed a surge in related research (Wen et al., 2023). They excel through strategies such as efficient attention mechanisms that optimize computational and memory requirements with sparse self-attention (Zhou et al., 2022a; Li et al., 2019), innovative model architectures featuring pyramidal attention modules for linear complexity (Liu et al., 2022), tailored attention mechanisms refining self-attention for specific time series patterns like Auto-Correlation (Wu et al., 2021), and the utilization of inherent time series data characteristics, incorporating techniques like patching and segment-based modelling to prioritize subsequence relationships over individual time points (Nie et al., 2023). Some work has also attempted to combine the Transformer with other network architectures, such as integrating it with Graph Neural Networks (GNNs) (Ng et al., 2022) or designing the Transformer in a U-Net structure (Madhusudhanan et al., 2022). However, transformer-based models often demonstrate limited temporal relation extraction capabilities and their redundancy in parameters can lead to increased susceptibility to overfitting, resulting in unstable prediction performance (Zeng et al., 2023).

In the realm of Multi-layer Perceptrons (MLPs), advancements in time series forecasting include utilizing interval and continuous sampling to partition input sequences and employing the Information Extraction Block (IEB) to extract patterns from tokens (Zhang et al., 2022). Unfortunately, such sampling may inevitably lead to information loss (Kreindler & Lumsden, 2016).

Compared to the previous literature, our VIPER can achieve more stable predictive performance with few parameters. This is achieved through the extraction and efficient utilization of intraperiod-variation and interperiod-variation patterns in the input sequence. Furthermore, VIPER can utilize longer input lengths, indicating that our VIPER is better at capturing temporal variation (Zeng et al., 2023).

# 3 METHODOLOGY

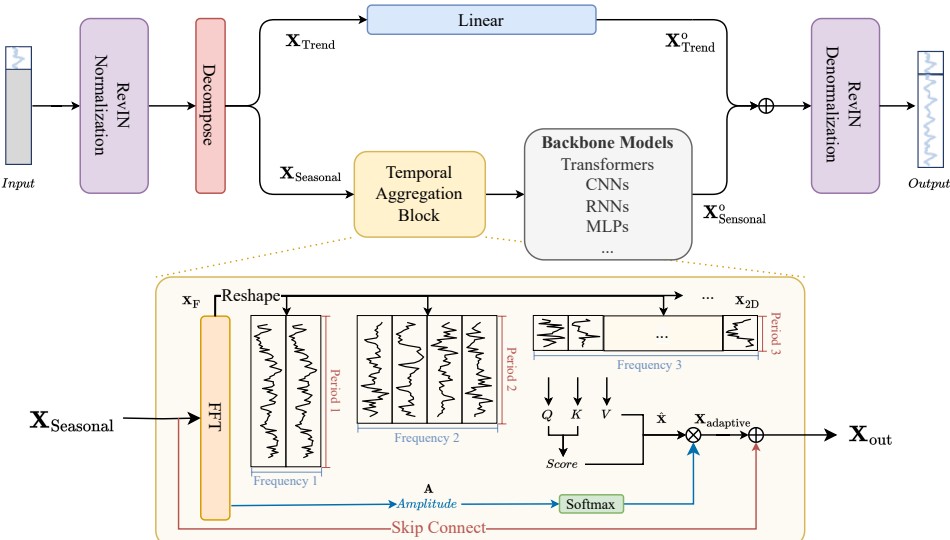

Figure 2: Framework of VIPER. After decomposing the original input time series into trend and seasonal parts, we employ a linear layer to directly project the trend part. Meanwhile, we model intraperiod- and interperiod-variation in the seasonal part using a temporal aggregation block, followed by projection using a backbone network. The two projected parts are then added together for forecasting. Additionally, RevIN is employed to ensure stability during the model training process.

The outline of the VIPER framework is depicted in Figure 2. Our strategy initiates by deconstructing the input sequence into its trend and seasonal components, referring to Figure 1 (b) for better understanding. Here, the seasonal parts demonstrate repetitive patterns, signifying periodicity. Concurrently, the trend component embodies the overall variances throughout the data sequence. Then, we employ distinct strategies for each component to forecast their future behaviour, the outcome of which is subsequently reincorporated after processing. Specifically, to process the seasonal component, we design a **Temporal Aggregation Block** to more dynamically and comprehensively capture patterns of both intraperiod- and interperiod-variation. This strategic approach enables us to obtain vibrant period representations for further robust and efficient forecasting.

## 3.1 TREND AND SEASONAL DECOMPOSITION

In the context of time series data, distribution shifts can occur when there are changes in the patterns, trends, or statistical properties of the data over different time periods, which leads to poor forecasting performance. To this end, we employed RevIN (Kim et al., 2022) to normalize and denormalize the input and output, respectively, to reduce the risk of model overfitting, enhance its generalization capability and stabilise the predictions. Specifically, we calculate the mean and variance of the input sequence, then normalize the input to zero mean and unit variance. Afterwards, we use this mean and variance to denormalize the obtained forecasting results.

Trends typically manifest as a sustained upward or downward movement in a time series over a certain period of time. They can obscure or confound the periodic components within a time series, thus affecting the observation and analysis of periodic patterns. To eliminate the interference of the trend part, we decompose the input $\mathbf{X}$ to $\mathbf{X}_{\text{Seasonal}}$ and $\mathbf{X}_{\text{Trend}}$, as shown in Figure 1 (b). For $\mathbf{X}_{\text{Trend}}$, we employ a moving average kernel with a stride of 25 to obtain the sliding average of the original sequence following Autoformer (Wu et al., 2021). Subsequently, $\mathbf{X}_{\text{Seasonal}}$ with repetitive patterns is obtained by subtracting the trend component from the original sequence $\mathbf{X}$.

**Trend Component** We employ a linear layer to project the trend component for trend forecasting according to $\mathbf{X}_{\text{Trend}}^{o} = \mathbf{W}_{\text{Trend}}\mathbf{X}_{\text{Trend}} + \mathbf{b}$. This approach, while simple, proves to be effective while consuming less memory and requiring fewer parameters. Thus, the result has a faster inference speed when compared to existing Transformer models.

**Seasonal Component** We introduce a novel Temporal Aggregation block (in Section 3.2), which enables dynamic fine-graining of feature representation from both intra-period and inter-period aspects. The extracted vibrant period representations are then fed into the subsequent backbone module (such as MLPs, CNNs, RNNs, and Transformers) for forecasting the seasonal component.

## 3.2 TEMPORAL AGGREGATION BLOCK

In the Temporal Aggregation Block, we first employ a multi-period transformation to convert the input time series into a series of 2D representations. Next, we utilize the Global Self-Attention block to model intraperiod-variations. Finally, we apply Adaptive Aggregation to learn dependencies among interperiod-variations.

**Multi-Period Transformation** To effectively represent the intraperiod-variation and interperiod-variation, the identification of underlying periods is crucial, especially when the input time series exhibits multi-periodic characteristics. Given seasonal component $\mathbf{X}_{\text{Seasonal}} \in \mathbb{R}^{N \times C}$, we first utilize the Fast Fourier Transform (FFT) to extract the top-k dominant frequencies, as illustrated in Figure 2.

$$\mathbf{X}_{\text{F}} = \text{FFT}(\mathbf{X}_{\text{Seasonal}}), \quad \mathbf{A} = \text{Topk}\left(\text{Avg}\left(\text{Amp}\left(\mathbf{X}_{\text{F}}\right)\right), k\right), \quad \mathbf{P} = \left\lceil \frac{seq\_len}{\mathbf{A}} \right\rceil, \quad (1)$$

where $\mathbf{X}_{\text{F}}$ is the sequence processed through FFT. $\mathbf{A}$ is the largest $k$ amplitudes. $\mathbf{P}$ stands for the period of the split sequence, $\text{Avg}$ denotes the average, $\text{Amp}$ represents the operation to get the amplification of the signal, and $seq\_len$ demonstrates the input sequence length. Subsequently, these extracted frequencies enable us to determine the corresponding periods.

We further transform $\mathbf{X}_{\text{Seasonal}}$ into a sequence of 2D representations to exploiting multi-period information, where one dimension corresponds to the length of a period and the other dimension represents the number of periods. This approach enables us to explicitly capture variations within and between periods in a 2D space, which reads:

$$\mathbf{X}_{2D} = \text{Reshape}\left(\mathbf{X}_{\text{Seasonal}}, \left[C, \frac{L}{\mathbf{P}}, \mathbf{P}\right]\right), \quad (2)$$

**Global Self-Attention Block** We use a global self-attention block to perform information aggregation on the obtained 2D representation. This design, utilizing a global self-attention mechanism, outperforms the 2D convolutional kernels employed by TimesNet in modelling comprehensive global relationships within the 2D representation from the same period division. Consequently, it excels in capturing intraperiod-variation more effectively. However, since the original self-attention requires the projection of inputs using three matrices $Q, K, V$, and due to the multi-period nature of the input sequence, this operation may introduce parameter redundancy, meaning that each period requires a specific set of $Q, K, V$. Therefore, we have removed this projection operation and refer to the remaining part as the Global Self-Attention Block, according to:

$$\hat{\mathbf{X}} = \text{Softmax}\left(\frac{\mathbf{X}_{2D}^T \mathbf{X}_{2D}}{\sqrt{(L/\mathbf{P})}}\right) \mathbf{X}_{2D}. \quad (3)$$

**Adaptive Aggregation** For the top-k selected periods, the amplitudes associated with each period to some extent reflect the weights of each period, the relationship among the amplitudes, and interperiod-variation. Therefore, we apply softmax to these top-k amplitudes and then perform weighted summation with the top-k 1D representations output by the Global Self-Attention Block. This process is called Adaptive Aggregation, which is used for modelling interperiod-variation. The result is then added together with $\mathbf{X}_{\text{Seasonal}}$ as the output of the Temporal Aggregation Block

$$\mathbf{X}_{\text{adaptive}} = \sum_{n=1}^{k} \text{Softmax}(\mathbf{A}_n)\hat{\mathbf{X}}, n \in \{1, \cdots, k\}, \quad \mathbf{X}_{\text{out}} = \mathbf{X}_{\text{Seasonal}} + \mathbf{X}_{\text{adaptive}}. \quad (4)$$

Table 1: The Statistics of the eight datasets used in our experiments.

| Datasets | ETTh1&2 | ETTm1&2 | Traffic | Electricity | Exchange-Rate | Weather |
|---|---|---|---|---|---|---|
| Variates | 7 | 7 | 862 | 321 | 8 | 21 |
| Timesteps | 17,420 | 69,680 | 17,544 | 26,304 | 7,588 | 52,696 |
| Granularity | 1 hour | 5 min | 1 hour | 1 hour | 1 day | 10 min |

## 4 EXPERIMENTAL RESULTS

### 4.1 DATASETS AND EVALUATION PROTOCOL

We selected eight widely-used real-world datasets that are multivariate time series, including Electricity Transformer Temperature (ETTh1, ETTh2, ETTm1, and ETTm2), Electricity, Traffic, Weather, and ExchangeRate (Wu et al., 2021). The characteristics of these datasets are shown in Table 1 (More can be found in Appendix). Following the previous works (Wu et al., 2023), we use Mean Squared Error (MSE) and Mean Absolute Error (MAE) as the core metrics for the evaluation.

### 4.2 BASELINES AND EXPERIMENTAL SETTING

We chose TimesNet (Wu et al., 2023), MICN (Wang et al., 2023), FEDformer (Zhou et al., 2022b), Autoformer (Wu et al., 2021), Informer (Zhou et al., 2022a), Pyraformer(Liu et al., 2022), Log-Trans (Li et al., 2019), as well as DLinear (Zeng et al., 2023), which achieved astonishing results with only two simple linear layers, as our baselines. All the models follow the same experimental setup with 4 different prediction lengths $T \in \{96, 192, 336, 720\}$. Our VIPER model uses an input length of $L = 720$ on all datasets except for the ExchangeRate. The authors of DLinear claimed that their best input length is $L = 336$, so we directly collected their performance results from their paper. MICN, on the other hand, had an optimal input length of 96; therefore, we also gathered their results directly from their paper. We directly collected performance metrics of other models from TimesNet (Wu et al., 2023), but we observed that all models in TimesNet have their input length set to $L = 96$. To avoid underestimating the performance of other methods, we rerun TimesNet, FEDformer, Autoformer and Informer for three different look-back windows $L \in \{96, 336, 720\}$, and always chose the best results to create strong and robust baselines following (Nie et al., 2023). In the experiments, we found that the remaining methods had the best input length of $L = 96$ on the ExchangeRate dataset. So we directly collected the results of the exchange rate dataset from TimesNet (Wu et al., 2023) as the performance of the remaining models. For a fair comparison, we also set the input sequence length of our VIPER model to $L = 96$ on the ExchangeRate dataset.

### 4.3 COMPARISION WITH THE STATE-OF-THE-ART METHODS

As evident from Table 2, the VIPER model significantly outperforms other state-of-the-art models across all the datasets, demonstrating its superior efficacy. Interestingly, VIPER and DLinear emerge as top performers across most datasets. This surprising result demonstrates their superiority over intricate, parameter-laden transformer-based models, MICN, and TimesNet. Despite the numerous TimesBlocks stacked in TimesNet leading to significant parameter redundancies, it fails to outperform our method. Moreover, the redundancy results of TimesNet in excessively high training costs compared to our approach. Similar to DLinear, VIPER also utilizes merely two fundamental linear layers, which leads to considerable reductions in both memory usage and training time. This implies that the strength of long-term time series forecasting doesn't necessarily rely on the sophistication of the model or extensive parameter tuning, but rather hinges upon a representation that accurately encapsulates the inherent semantic information, such as trend and seasonal DLinear and intraperiod-variation and interperiod-variation patterns in VIPER, of the time series.

### 4.4 INPUT LENGTH BOTTLENECK ANALYSIS

The power of a better representation—namely, one that effectively retains critical historical information while proficiently filtering out noise from the original sequence—is further assessed through extensive experimentation. We carefully chose illustrative works from various facets of time-series forecasting, including LSTM (Hochreiter & Schmidhuber, 1997) (representing RNNs), Informer (Zhou et al., 2022a) and Pyraformer (Liu et al., 2022) (two noteworthy transformer-based models), SCINet (LIU et al., 2022) (an avant-garde model utilizing Temporal Convolutional Network), and LightTS (Zhang et al., 2022) and DLinear (Zeng et al., 2023) (two MLP-based models

Table 2: Multivariate long-term forecasting result comparison. We use prediction lengths $T \in \{96, 192, 336, 720\}$. The best results are in **bold** and the second bests are underlined.

| | Metric | VIPER (Ours) MSE MAE | MICN (2023) MSE MAE | TimesNet (2023) MSE MAE | DLinear (2023) MSE MAE | FEDformer (2022b) MSE MAE | Autoformer (2021) MSE MAE | Informer (2022a) MSE MAE | Pyraformer (2022) MSE MAE | LogTrans (2019) MSE MAE |
|---|---|---|---|---|---|---|---|---|---|---|
| ETTm1 | 96 | 0.307 0.349 | 0.316 0.362 | 0.338 0.375 | **0.299 0.343** | 0.326 0.390 | 0.510 0.492 | 0.626 0.560 | 0.543 0.510 | 0.600 0.546 |
| | 192 | 0.337 0.367 | 0.363 0.390 | 0.374 0.387 | **0.335 0.365** | 0.365 0.415 | 0.514 0.495 | 0.725 0.619 | 0.557 0.537 | 0.837 0.700 |
| | 336 | **0.366 0.384** | 0.408 0.426 | 0.410 0.411 | 0.369 0.386 | 0.392 0.425 | 0.510 0.492 | 1.005 0.741 | 0.754 0.655 | 1.124 0.832 |
| | 720 | **0.416 0.412** | 0.481 0.476 | 0.478 0.450 | 0.425 0.421 | 0.446 0.458 | 0.527 0.493 | 1.133 0.845 | 0.908 0.724 | 1.153 0.820 |
| ETTm2 | 96 | **0.161 0.251** | 0.179 0.275 | 0.187 0.267 | 0.167 0.260 | 0.180 0.271 | 0.205 0.293 | 0.355 0.462 | 0.435 0.507 | 0.768 0.642 |
| | 192 | **0.215 0.289** | 0.307 0.376 | 0.249 0.309 | 0.224 0.303 | 0.252 0.318 | 0.278 0.336 | 0.595 0.586 | 0.730 0.673 | 0.989 0.757 |
| | 336 | **0.267 0.325** | 0.325 0.388 | 0.321 0.351 | 0.281 0.342 | 0.324 0.364 | 0.343 0.379 | 1.270 0.871 | 1.201 0.845 | 1.334 0.872 |
| | 720 | **0.350 0.377** | 0.502 0.490 | 0.408 0.403 | 0.397 0.421 | 0.410 0.420 | 0.414 0.419 | 3.001 1.267 | 3.625 1.451 | 3.048 1.328 |
| ETTh1 | 96 | **0.368 0.398** | 0.421 0.431 | 0.384 0.402 | 0.375 0.399 | 0.376 0.415 | 0.435 0.446 | 0.941 0.769 | 0.664 0.612 | 0.878 0.740 |
| | 192 | **0.403** 0.419 | 0.474 0.487 | 0.436 0.429 | 0.405 **0.416** | 0.423 0.446 | 0.456 0.457 | 1.007 0.786 | 0.790 0.681 | 1.037 0.824 |
| | 336 | **0.423 0.436** | 0.569 0.551 | 0.491 0.469 | 0.439 0.443 | 0.444 0.462 | 0.486 0.487 | 1.038 0.784 | 0.891 0.738 | 1.238 0.932 |
| | 720 | **0.426 0.455** | 0.770 0.672 | 0.521 0.500 | 0.472 0.490 | 0.469 0.492 | 0.515 0.517 | 1.144 0.857 | 0.963 0.782 | 1.135 0.852 |
| ETTh2 | 96 | **0.268 0.332** | 0.299 0.364 | 0.340 0.374 | 0.289 0.353 | 0.332 0.374 | 0.332 0.368 | 1.549 0.952 | 0.645 0.597 | 2.116 1.197 |
| | 192 | **0.329 0.372** | 0.441 0.454 | 0.402 0.414 | 0.383 0.418 | 0.407 0.446 | 0.426 0.434 | 3.792 1.542 | 0.788 0.683 | 4.315 1.635 |
| | 336 | **0.345 0.391** | 0.654 0.567 | 0.407 0.446 | 0.448 0.465 | 0.400 0.447 | 0.477 0.479 | 4.215 1.642 | 0.907 0.747 | 1.124 1.604 |
| | 720 | **0.376 0.422** | 0.956 0.716 | 0.404 0.443 | 0.605 0.551 | 0.412 0.469 | 0.453 0.490 | 3.656 1.619 | 0.963 0.783 | 3.188 1.540 |
| Electricity | 96 | **0.134 0.230** | 0.310 0.398 | 0.168 0.272 | 0.140 0.237 | 0.186 0.302 | 0.196 0.313 | 0.304 0.393 | 0.386 0.449 | 0.258 0.357 |
| | 192 | **0.148 0.243** | 0.300 0.394 | 0.184 0.289 | 0.153 0.249 | 0.197 0.311 | 0.211 0.324 | 0.327 0.417 | 0.386 0.443 | 0.266 0.368 |
| | 336 | **0.164 0.259** | 0.323 0.413 | 0.198 0.300 | 0.169 0.267 | 0.213 0.328 | 0.214 0.327 | 0.333 0.422 | 0.378 0.443 | 0.280 0.380 |
| | 720 | 0.204 **0.292** | 0.364 0.449 | 0.220 0.320 | 0.203 0.301 | 0.233 0.344 | 0.236 0.342 | 0.351 0.427 | 0.376 0.445 | 0.283 0.376 |
| Weather | 96 | 0.168 0.222 | **0.161** 0.229 | 0.172 **0.220** | 0.176 0.237 | 0.238 0.314 | 0.249 0.329 | 0.354 0.405 | 0.896 0.556 | 0.458 0.490 |
| | 192 | **0.212 0.259** | 0.220 0.281 | 0.219 0.261 | 0.220 0.282 | 0.275 0.329 | 0.325 0.370 | 0.419 0.434 | 0.622 0.624 | 0.658 0.589 |
| | 336 | **0.259 0.296** | 0.278 0.331 | 0.280 0.306 | 0.265 0.319 | 0.339 0.377 | 0.351 0.391 | 0.583 0.543 | 0.739 0.753 | 0.797 0.652 |
| | 720 | 0.319 **0.338** | **0.311** 0.356 | 0.365 0.359 | 0.323 0.362 | 0.389 0.409 | 0.415 0.426 | 0.916 0.705 | 1.004 0.934 | 0.869 0.675 |
| Traffic | 96 | **0.388 0.272** | 0.519 0.309 | 0.593 0.321 | 0.410 0.282 | 0.576 0.359 | 0.597 0.371 | 0.733 0.410 | 2.085 0.468 | 0.684 0.384 |
| | 192 | **0.397 0.273** | 0.537 0.315 | 0.585 0.321 | 0.423 0.287 | 0.610 0.380 | 0.607 0.382 | 0.777 0.435 | 0.867 0.467 | 0.685 0.390 |
| | 336 | **0.411 0.279** | 0.534 0.313 | 0.621 0.336 | 0.436 0.296 | 0.608 0.375 | 0.623 0.387 | 0.776 0.434 | 0.869 0.469 | 0.734 0.408 |
| | 720 | **0.450 0.301** | 0.577 0.325 | 0.637 0.345 | 0.466 0.315 | 0.621 0.375 | 0.639 0.395 | 0.827 0.466 | 0.881 0.473 | 0.717 0.396 |
| Exchange | 96 | **0.081 0.197** | 0.102 0.235 | 0.107 0.234 | 0.088 0.218 | 0.148 0.278 | 0.197 0.323 | 0.847 0.752 | 1.748 1.105 | 0.968 0.812 |
| | 192 | **0.172 0.293** | 0.172 0.316 | 0.226 0.344 | 0.176 0.315 | 0.271 0.380 | 0.300 0.369 | 1.204 0.895 | 1.874 1.151 | 1.040 0.851 |
| | 336 | 0.319 **0.406** | 0.272 0.407 | 0.367 0.448 | 0.313 0.427 | 0.460 0.500 | 0.509 0.524 | 1.672 1.036 | 1.943 1.172 | 1.659 1.081 |
| | 720 | 0.854 0.689 | **0.714 0.658** | 0.964 0.746 | 0.839 0.695 | 1.195 0.841 | 1.447 0.941 | 2.478 1.310 | 2.085 1.206 | 1.941 1.127 |

comparative to state-of-the-art transformer models), and TimesNet (Wu et al., 2023). For the experiments, medium-sized and longer time-step datasets—ETTh2, ETTm2, and Weather—were selected to mitigate overfitting. The prediction length was deliberately set to $T = 720$, with input lengths varied between $L \in \{96, 192, 336, 720, 960\}$, aiming to gain a more comprehensive view of the experimental outcomes. MAE was chosen as the evaluation metric.

Figure 3 reveals a phenomenon we refer to as the 'Input Length Bottleneck Problem'. There is an implicit expectation that increasing the input length with more information involved would proportionally enhance the performance. However, contrasting results are observed in the other methods. As the input sequence length elongated, the performance of most other models either descended or demonstrated an initial surge, subsequently followed by deterioration in performance. Such a pattern implies that these models struggle significantly with effectively capitalizing on the surplus input information, a similar finding echoed in Zeng et al. (2023).

Surprisingly, both VIPER and DLinear perform admirably when dealing with extended input sequences, particularly noticeable on datasets such as Weather and ETTh2. However, DLinear encountered difficulties when it came to the ETTm2 dataset, unlike VIPER, which exhibited robust performance across all three datasets consistently.

Furthermore, it's worth highlighting that TimesNet experienced a degradation in performance as the input length increased. This accentuates the superiority of our proposed global self-attention method compared to the utilization of 2D convolution kernels along with local self-attention windows for feature extraction. Moreover, this reveals that employing the multi-period transformations on the seasonal component potentially performs better than on the raw sequence.

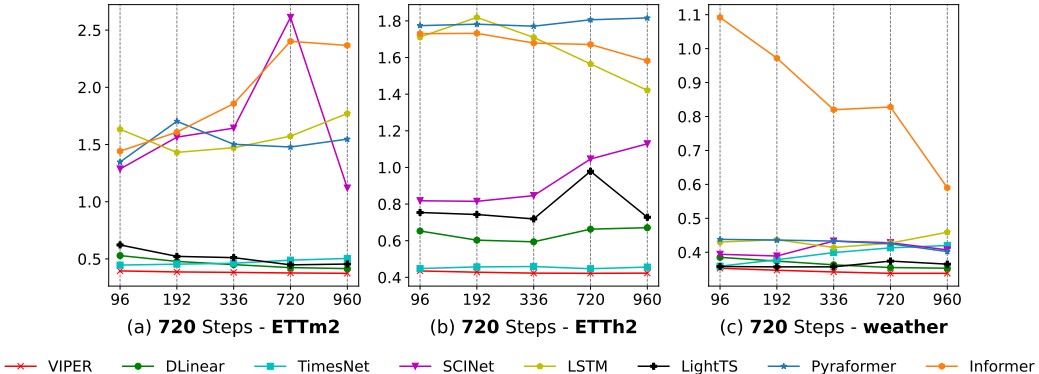

Figure 3: Different MAE performances (Y-Axis) vary on models and input lengths (X-Axis). Our model reached the best with steadily increasing performance with the input length increases.

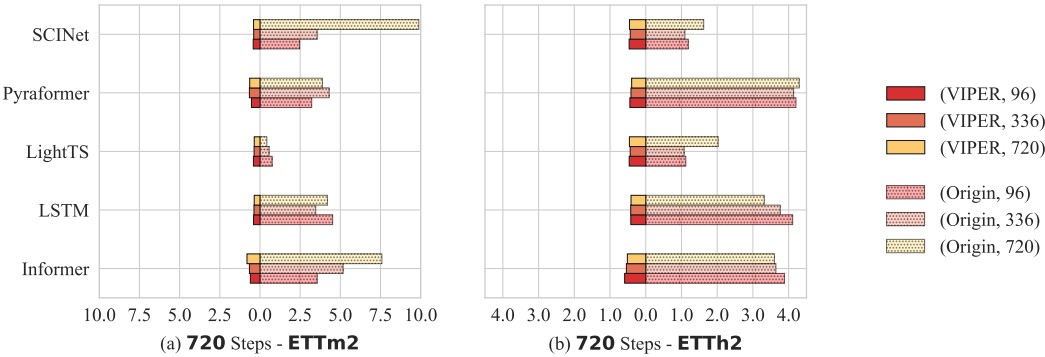

Figure 4: The MSE results (X-Axis) of various models with and without VIPER of long-term forecasting (T=720) on the ETTm2 (a) and ETTh2 (b) datasets. The shorter, the better.

### 4.5 BOOSTING PERFORMANCE OF OTHER MODELS

To further validate our perspective that improved representations are crucial for addressing time series prediction challenges, and showcase the versatility of our VIPER framework, we conducted comprehensive experiments. These involved sequentially substituting the backbone in VIPER with an array of architectures, including LSTM (RNNs), Informer and Pyraformer (both transformers), SCINet (CNNs), and LightTS (MLPs). We selected ETTh2 and ETTm2 datasets for our experiments. In these tests, we set the prediction length at $T = 720$. To examine how our method assists models in overcoming the input length bottleneck problem, we preset the input length to $L \in \{96, 336, 720\}$. The evaluation metric was MSE.

As illustrated in Figure 4, our approach significantly enhanced the performance of all models in MSE metric on both datasets. SCINet, Informer, Pyraformer, and LSTM experienced remarkable improvement using our method, while LightTS also saw substantial enhancement. These results suggest that superior input feature representation, rather than the specific model type, is the key to successful long-term time series forecasting.

### 4.6 ABLATION STUDIES

**Model Structure**  To explore the individual contribution of each component (Temporal Aggregation Block and RevIN) of VIPER, we conducted ablation studies on five datasets. According to the results shown in Table 3, it is evident that our Temporal Aggregation Block further enhanced the performance of DLinear. The addition of RevIN also successfully stabilized the training process of the model, resulting in an obvious performance improvement.

**Comparative Experiment about Temporal Aggregation Block and TimesBlock**  We conducted

Table 3: Ablations on model architecture. We defined 'DL' as the original model i.e., DLinear, without introducing anything, and defined 'TB' as introducing Temporal Aggregation Block on the basis of DLinear, and 'TB+RevIN' as adding RevIN on the basis of 'TB' to stabilize the training process. Input sequence length of all datasets are set to L=96, and prediction length are set to T=720.

| Datasets | ETTm1 | | ETTm2 | | ETTh2 | | ETTh1 | | Traffic | | Avg | |
|---|---|---|---|---|---|---|---|---|---|---|---|---|
| Metrics | MSE | MAE | MSE | MAE | MSE | MAE | MSE | MAE | MSE | MAE | MSE | MAE |
| DL | 0.472 | 0.451 | 0.559 | 0.529 | 0.831 | 0.657 | 0.517 | 0.513 | 0.674 | 0.420 | 0.562 | 0.500 |
| +TB | **0.471** | 0.450 | 0.539 | 0.514 | 0.803 | 0.645 | 0.506 | 0.504 | 0.646 | 0.397 | 0.541 | 0.487 |
| +TB+RevIN | 0.478 | **0.446** | **0.406** | **0.395** | **0.415** | **0.435** | **0.476** | **0.467** | **0.643** | **0.383** | **0.486** | **0.437** |

Table 4: Extensive Experiment of Temporal Aggregation Block and TimesBlock (Wu et al., 2023), the basic component of TimesNet, on ETTh2, ETTm2. TB means directly apply a TimesBlock to raw data and then using a linear layer to get the forecasting result. TB+De means we first decompose the raw data to seasonal and trend, then apply TimesBlock on seasonal part, which equals to supplement the Temporal Aggregation Block in VIPER to TimesBlock.

| Datasets | L | VIPER | | TB + De | | TB | |
|---|---|---|---|---|---|---|---|
| | | MSE | MAE | MSE | MAE | MSE | MAE |
| ETTh2 | 96 | **0.415** | **0.435** | 0.497 | 0.480 | 0.545 | 0.514 |
| | 192 | **0.399** | **0.428** | 0.493 | 0.485 | 0.508 | 0.497 |
| | 336 | **0.384** | **0.423** | 0.494 | 0.508 | 0.510 | 0.510 |
| | 720 | **0.376** | **0.422** | 0.566 | 0.563 | 0.568 | 0.572 |
| ETTm2 | 96 | **0.406** | **0.395** | 0.424 | 0.413 | 0.443 | 0.419 |
| | 192 | **0.381** | **0.386** | 0.435 | 0.420 | 0.460 | 0.432 |
| | 336 | **0.366** | **0.382** | 0.445 | 0.432 | 0.455 | 0.443 |
| | 720 | **0.350** | **0.377** | 0.448 | 0.441 | 0.485 | 0.463 |

experiments to show the advantages of decomposing the original time series and using global attention for intraperiod-variation and interperiod-variation learning. We compared TimesBlock, a key component of TimesNet that employs 2D convolutional kernels for feature extraction, with our VIPER, which utilizes global attention in the Temporal Aggregation Block.

Results in Table 4 on the ETTh2 and ETTm2 datasets reveal significant performance improvement for TimesBlock with the decomposition strategy. This highlights the effectiveness of decomposing data before applying 2D feature transformations to the seasonal part. Notably, both TimesBlock methods show performance degradation with longer input sequences, while our VIPER, thanks to global attention in the Temporal Aggregation Block, achieves positive enhancements.

In general, a strong TSF model, with robust temporal relation extraction capabilities, benefits from larger look-back window sizes. This underscores the limitations of local attention in 2D convolutional kernels for extracting and utilizing temporal patterns, particularly in longer sequences. In contrast, our design with global attention excels in extracting temporal information, enabling longer output windows. More visualization and analysis can be found in Appendix A.2.

## 5 CONCLUSION AND FUTURE WORK

We introduce the VIPER framework, leveraging the multi-period characteristics of real-world data. It efficiently disentangles and eliminates interference from trend components, allowing for focused modelling of dependencies within the seasonal component. VIPER enables the extraction of both intraperiod-variation and interperiod-variation, resulting in more robust and efficient forecasting outcomes. Furthermore, VIPER has highlighted the advantages of employing longer input sequences, addressing the well-known Input Length Bottleneck Problem. Notably, VIPER has demonstrated exceptional predictive performance while requiring a remarkably low number of parameters, showcasing its potential for wider application. In the future, we aim to extend VIPER's applications to domains such as classification, imputation, anomaly detection, and beyond.

## 6 REPRODUCIBILITY STATEMENT

To foster reproducibility, we make our code available in supplementary materials and will public it online after acceptance. We give details on our experimental protocol in Appendix A.1.

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

# A    APPENDIX

## A.1    EXPERIMENTAL DETAILS

### A.1.1    DATASET DETAILS

We elaborate on the datasets employed in this study with the following details.

- **ETT Dataset** (Zhou et al., 2022a) comprises two sub-datasets: **ETTh** and **ETTm**, which were collected from electricity transformers. Data were recorded at 15-minute and 1-hour intervals for ETTm and ETTh, respectively, spanning from July 2016 to July 2018.

- **ExchangeRate Dataset** (Lai et al., 2018) provides daily exchange rates for eight different countries, spanning from 1990 to 2016.

- **Electricity1 Dataset**[1] encompasses the electricity consumption data of 321 customers, recorded on an hourly basis, covering the period from 2012 to 2014.

- **Traffic Dataset**[2] consists of hourly data from the California Department of Transportation. It describes road occupancy rates measured by various sensors on San Francisco Bay area freeways.

- **Weather Dataset**[3] contains records of 21 meteorological indicators, updated every 10 minutes throughout the entire year of 2020.

### A.1.2    IMPLEMENTATION DETAILS AND MODEL PARAMETERS

We trained our VIPER model using the L2 loss function and employed the ADAM optimizer with an initial learning rate of $2.5e-5$. The batch size was set to 32, and we initialized the random seed as 2021. We also configured the hyperparameter top-k to 5. Additionally, we set the optimal input length to $L = 720$. During the training process, we incorporated an early stopping mechanism, which would halt training after three epochs if no significant reduction in loss was observed on the validation set. For evaluation purposes, we used two key performance metrics: the mean square error (MSE) and the mean absolute error (MAE). Our implementation was carried out in PyTorch and executed on an NVIDIA V100 32GB GPU. All the code and experimental details will be public in the future. We also attach the code in a supplementary file for your reference and run the demo.

## A.2    RESULT VISUALIZATION

### A.2.1    VARYING LOOK-BACK WINDOW

In principle, extending the look-back window increases the receptive field, leading to a potential improvement in forecasting performance. A robust Time Series Forecasting (TSF) model equipped with a strong temporal relation extraction capability should yield improved results with larger look-back window sizes. As demonstrated in Figure 5, Our VIPER model consistently and effectively diminishes both MSE and MAE scores as the receptive field expands, affirming its capacity to leverage longer look-back windows and superior temporal relation extraction capabilities.

### A.2.2    DECOMPOSITION RESULTS AND VISUALIZATION OF ALL DATASETS

The presence of a trend may potentially mask or confuse the periodic patterns in a time series, making it difficult to identify the underlying cyclic variations in the raw sequence, thereby hindering our ability to perform multi-period transformations. We can see in Figure 6 that, after decomposition, the seasonal component exhibits a more pronounced periodicity compared to the original input sequence. Therefore, it is more suitable for performing multi-period feature transformations.

---

[1]https://archive.ics.uci.edu/ml/datasets/ElectricityLoadDiagrams20112014
[2]https://pems.dot.ca.gov/
[3]https://www.bgc-jena.mpg.de/wetter/

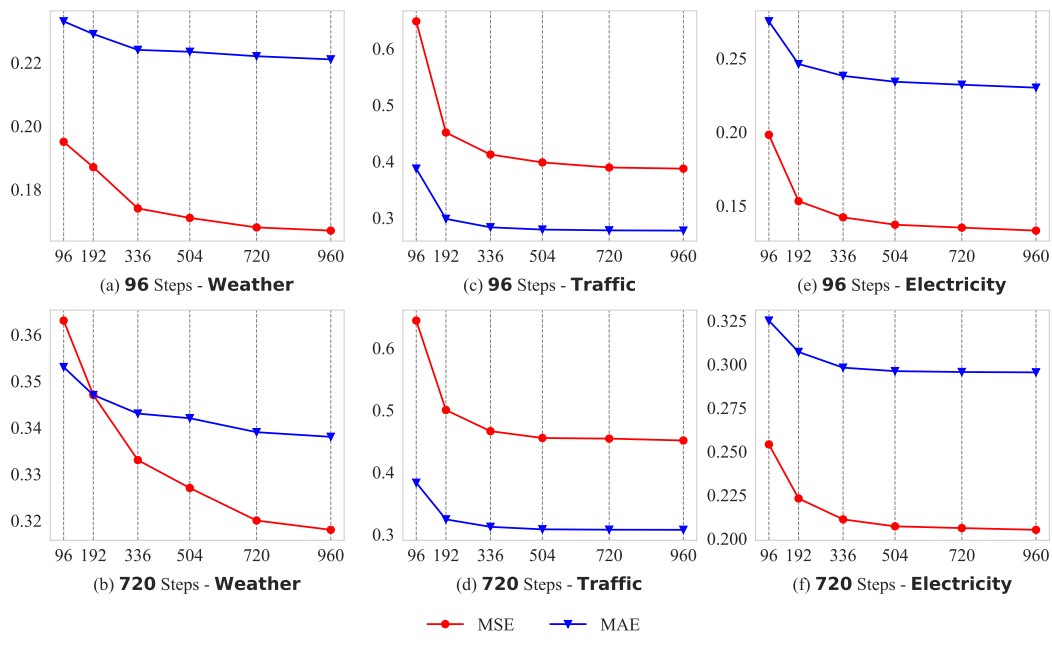

Figure 5: Forecasting performance (MSE and MAE) of VIPER with varying look-back windows on 3 datasets: Electricity, Traffic, and Weather. The look-back windows are selected to be L = 96, 192, 336, 504, 720, 960, and the prediction horizons are T = 96, 720.

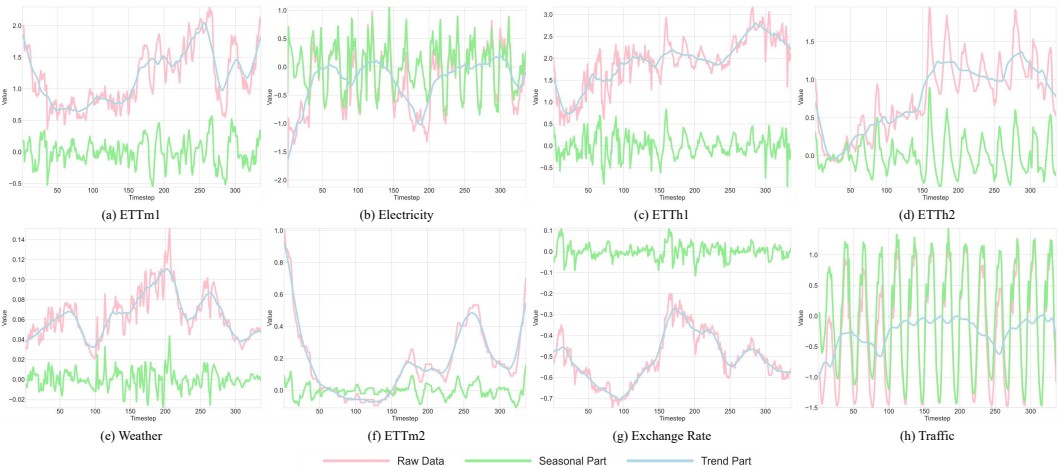

Figure 6: The decomposition and visualization results of the eight datasets used in the experiment. We take the time series of the last dimension of each dataset with a length L=336 for analysis. Using the same decomposition strategy as Autoformer (Wu et al., 2021), we decompose each time series into trend and seasonal parts.

## A.3 ATTENTION VISUALIZATION

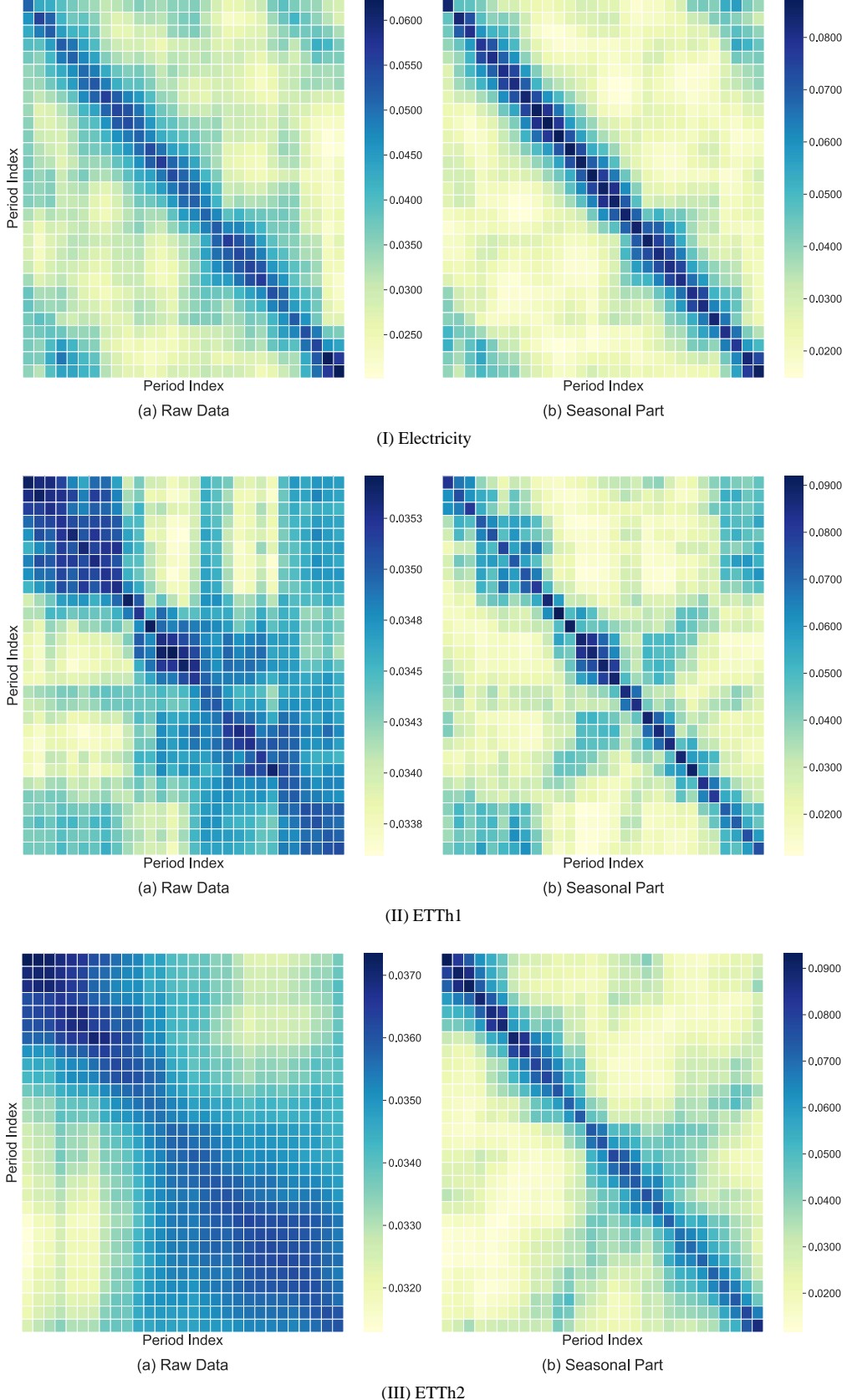

(a) Raw Data        (b) Seasonal Part

(I) Electricity

(a) Raw Data        (b) Seasonal Part

(II) ETTh1

(a) Raw Data        (b) Seasonal Part

(III) ETTh2

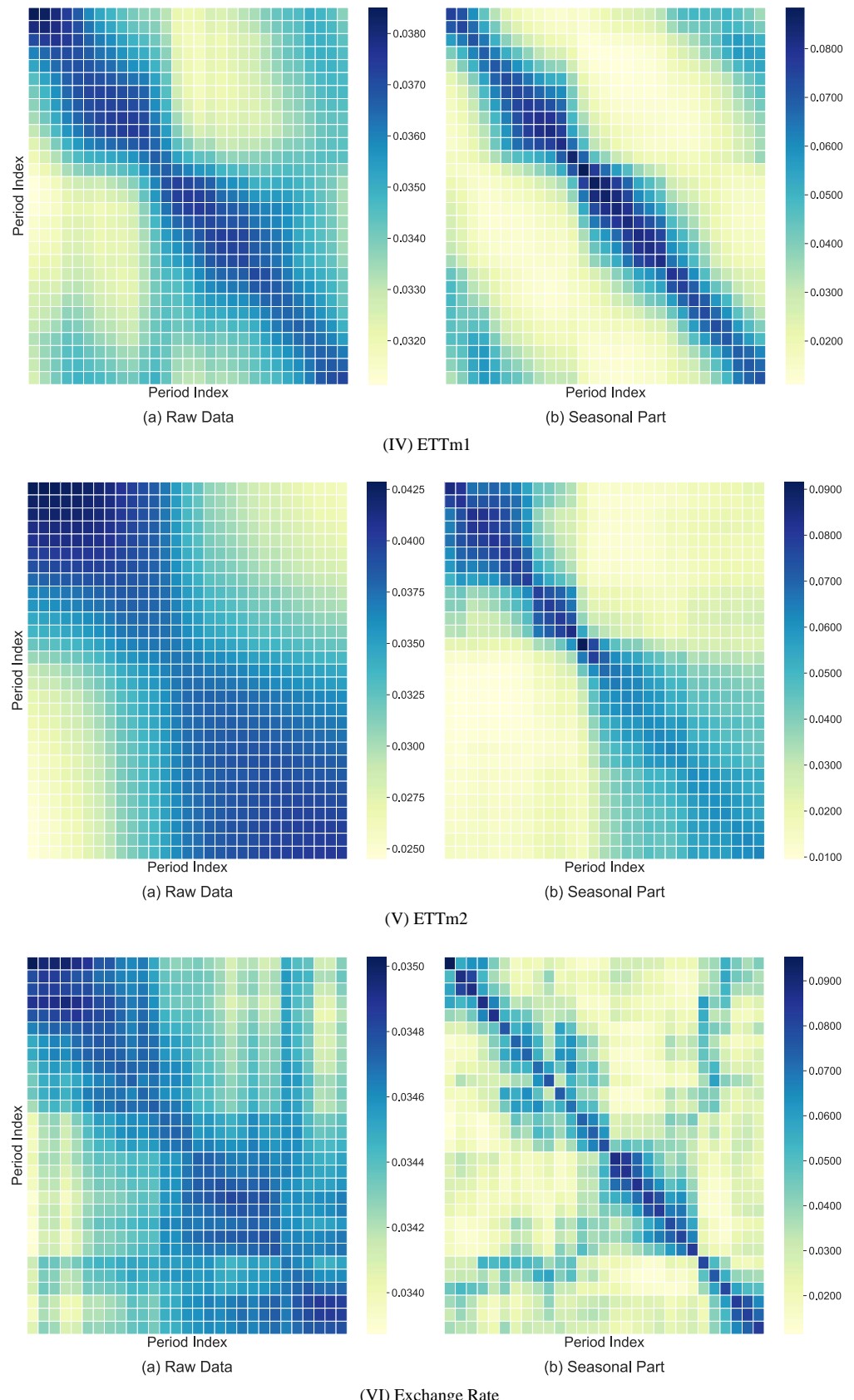

(a) Raw Data      (b) Seasonal Part

(IV) ETTm1

(a) Raw Data      (b) Seasonal Part

(V) ETTm2

(a) Raw Data      (b) Seasonal Part

(VI) Exchange Rate

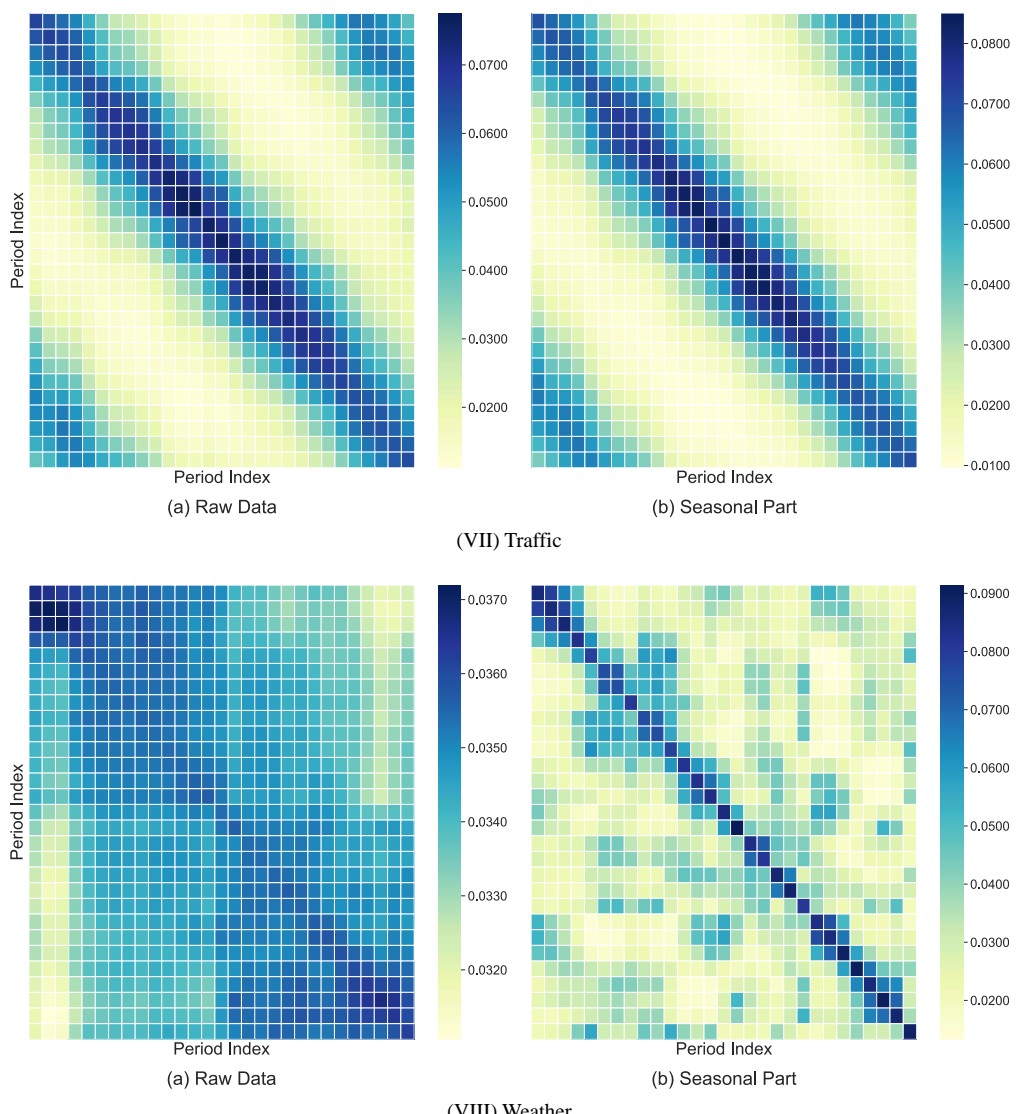

Figure 7: Visualization of the attention map in the Raw-Attention Block within the Temporal Aggregation Block for different datasets. To achieve more conspicuous visualization results, we chose the time series of the last dimension for each dataset. The input length for all dataset time series was set to $L = 336$. We selected the most suitable period from the top-k period extracted from the raw input series and partitioned the $input$ (raw input sequence or its seasonal part) accordingly. Specifically, for an input sequence denoted as $input : [seq\_len]$, we partitioned it into $feature : [seq\_len/period, period]$, and then computed its self-attention score. Finally, the attention scores were visualized.

It is readily apparent that when a time series exhibits conspicuous periodicity, the raw sequence and its seasonal component share a strikingly high degree of similarity in their attention maps. For instance, in the context of time series data, such as Traffic and Electricity, these datasets' time sequences, when examined in the decomposition diagrams in Figure 6 (b)(d), unmistakably reveal that both the seasonal part and the original input sequence possess distinctly marked and closely aligned periodic characteristics. Consequently, their attention visualization maps in Figure 7 (I)(VII) for the original sequence and its seasonal component exhibit conspicuously analogous patterns. However, in contrast to the original sequence, the seasonal component demonstrates an even more robust level of correlation. Consequently, within the attention visualization map of the seasonal component, attention tends to converge more profoundly. This convergence is discernible through higher values in the heatmap, signifying an augmented similarity between two highly corresponding periods

within the original sequence. Conversely, periods within the original sequence that lack resemblance display diminished similarity within the attention visualization graph of the seasonal part.

In scenarios where the original time series lacks evident periodicity, as observed in datasets such as ETTh1&2, ETTm1&2, and ExchangeRate, it becomes apparent, as depicted in Figure 6 (a)(c)(d)(f)(g), that the original input sequence does not manifest clear periodicity. Consequently, their attention map for the original sequence in Figure 7 (II)(III)(IV)(V)(VI) tend to exhibit a greater degree of divergence and fail to reveal discernible patterns. This observation suggests that subjecting such original sequences to multi-period transformations yields features with diminished semantic information. Nonetheless, following decomposition, the seasonal component consistently manifests more pronounced periodicity. Consequently, the attention heatmap for the seasonal component reveals a heightened convergence. Each period not only captures neighboring periods but also extends its attention to remote, albeit similarly patterned, periods.

