# OpenReview forum: "VIPER: Vibrant Period Representation for Robust and Efficient Time Series Forecasting"
_ICLR.cc/2024/Conference — Submitted to ICLR 2024_

### Official Review · Reviewer_PNtL · 2023-10-27

**Soundness:** 2 fair
**Presentation:** 2 fair
**Contribution:** 2 fair
**Rating:** 3
**Confidence:** 5

**Summary:**

This paper proposes a framework that decomposes time series data into trends and seasonal data to solve the time series forecasting problem.

**Strengths:**

1. The authors propose a framework to get good performance compared with other methods on some real datasets.

**Weaknesses:**

1. The authors highlight that they decompose the time series data into two variants, but I didn't get how to decompose the data. I believe the author should give some technical details about this important part.
2. Many previous works have already decomposed the time series data into trends and seasonal data to solve the TSF problem. What sets this paper apart from other similar methods? If it's about the efficiency of model training and prediction (I think the authors highlight it in paragraph 2 of the introduction), it wasn't emphasized in the experimental section.
3. Does this method assume that all the time series data has periodic patterns? The author emphasizes the distribution shift of the data, which contradicts periodicity. If the normalized data exhibits periodicity, the author didn't conduct an analysis in the experiment.

**Questions:**

please see the weaknesses

---

### Official Review · Reviewer_juBb · 2023-10-31

**Soundness:** 2 fair
**Presentation:** 2 fair
**Contribution:** 2 fair
**Rating:** 3
**Confidence:** 4

**Summary:**

This paper presents a plug-in period representation block designed for time series forecasting. The proposed representation block first reshapes time series based on their (multiple) periods, followed by inter-period filtering, referred to as the global self-attention block in the paper, to generate the updated series. These updated series are subsequently fed into the forecasting model. Experimental results show that it is competing, and even slightly surpassing SOTA methods.

**Strengths:**

1.	The proposed period representation block is reasonably designed for processing periodical time series.
2.	The proposed period representation block is versatile and can seamlessly integrate into various forecasting models.

**Weaknesses:**

1.	The overall novelty of this paper is not high. The presentation block is actually a simple seasonal filtering algorithm with Transformer as a shell.
2.	The experiments are flawed. Please check the Questions part for detailed discussion.
3.	The presentation of this paper can be further improved. Some writings lack clarify, and there are some formula confusions and other issues (please check the Question part).

**Questions:**

1. What is the purpose of the "Avg" in Equation (1)?
2. The representation of Equation (3) needs clarification. Which axis is the attention operation performed on?
3. The authors mention the importance of intraperiod-variation and interperiod-variation in time series representation learning. However, it is not clear how the proposed representation block captures these variations separately.
4. The results of TimesNet in Table 2 appear inconsistent with the original paper. In [1], TimesNet outperformed DLinear, while this paper concludes the opposite.
5. The reported Avg results in Table 3 are incorrect. Besides, the proposed block only yields a slight 2.9% improvement in MSE according to my calculation.

[1] Haixu Wu, et, al. TimesNet: Temporal 2D-Variation Modeling for General Time Serie Analysis. ICLR 2023.

---

### Official Review · Reviewer_NQNh · 2023-11-01

**Soundness:** 2 fair
**Presentation:** 4 excellent
**Contribution:** 2 fair
**Rating:** 3
**Confidence:** 4

**Summary:**

This article presents a new architecture for time series forecasting. This architecture is based on decomposing the signal into seasonal and trend components. Regarding the trend, a linear model is used. For the seasonal component, Fast Fourier Transform (FFT) is initially applied. The most important frequencies are retained, and the transformed signal is then passed through a block called the Temporal Aggregation Block. This block employs global attention to discover intra-period patterns, followed by weighting based on the importance of frequencies. Finally, a backbone (CNN, RNN, etc.) is used to obtain the final representation of the seasonal signal. This seasonal signal and the one predicted by the trend are then added together to obtain the final prediction. Experiments are conducted on the usual baselines and datasets in the field.

**Strengths:**

* The article is well-written, easy to read, the literature review is substantial and well-organized.
* The conducted experiments are appropriate for the forecasting problem.
* An ablation study is performed to demonstrate the method's significance.
* The appendices contain interesting experiments on attention visualization.

**Weaknesses:**

* In my opinion, the main novelty of the article is the temporal aggregation block - the decomposition between trend and season is quite standard, as is the use of FFT. This block consists of global attention and frequency weighting. To me, this represents a very minor innovation compared to what already exists.

* The major issue with this article concerns the absence of PatchTST [Nie et al. 2023] in the baselines. This work is mentioned 2 or 3 times in the paper, including the experimental section, so the authors are aware of its existence, but surprisingly, the results of PatchTST are not included in the result table. However, the results of the latter are almost always better than the proposed approach. This may not necessarily be a problem in itself - there could be other advantages to the proposed architecture. However, the deliberate omission of this comparison does not adhere to good scientific practices.

**Questions:**

It is usual in forecasting articles to include experiments on incomplete data, where a portion of the data is masked. Do you have such results?

---

### Meta-Review · Area_Chair_qHHE · 2023-12-06

**Metareview:**

Reviewers highlighted various concerns, notably the lack of baseline models. The absence of author feedback hindered addressing critical questions, impeding the submission's improvement to meet conference standards.

**Justification For Why Not Higher Score:**

The absence of author feedback hindered addressing critical questions, impeding the submission's improvement to meet conference standards.

**Justification For Why Not Lower Score:**

n/a

---

### Decision · Program_Chairs · 2024-01-16

Reject